# Alginate Core-Shell Capsules Production through Coextrusion Methods: Principles and Technologies

**DOI:** 10.3390/md21040235

**Published:** 2023-04-11

**Authors:** Chanez Bennacef, Sylvie Desobry-Banon, Laurent Probst, Stéphane Desobry

**Affiliations:** 1Laboratoire D’ingenierie des Biomolécules (LIBio), ENSAIA-Université de Lorraine, 2 Avenue de la Forêt de Haye, CEDEX, BP 20163, 54505 Vandœuvre-lès-Nancy, France; chanez.bennacef@univ-lorraine.fr (C.B.);; 2Cookal SAS Company, 19 Avenue de la Meurthe, 54320 Nancy, France

**Keywords:** coextrusion, alginate, encapsulation, marine polymer, hydrogel, dripping, jetting, centrifugal coextrusion, EHD

## Abstract

This paper provides an overview of coextrusion methods for encapsulation. Encapsulation involves the coating or entrapment of a core material such as food ingredients, enzymes, cells, or bioactives. Encapsulation can help compounds add to other matrices, stabilize compounds during storage, or enable controlled delivery. This review explores the principal l coextrusion methods available that can be used to produce core-shell capsules through the use of coaxial nozzles. Four methods for encapsulation by coextrusion are examined in detail, including dripping, jet cutting, centrifugal, and electrohydrodynamic systems. The targeted capsule size determines the appropriate parameters for each method. Coextrusion technology is a promising encapsulation technique able to generate core-shell capsules in a controlled manner, which can be applied to cosmetic, food, pharmaceutical, agriculture, and textile industries. Coextrusion is an excellent way to preserve active molecules and present a significant economic interest.

## 1. Introduction

Encapsulation technologies began back in 1872 with Samuel Percy’s conception of spray drying. By 1875, Cains suggested a coating technique and described the basic fundaments of coating and enrobing still used today, in particular with bottom spray fluid-bed coating. Later, in 1957, Green and Schleicher invented a micrometric oil-containing capsule that would later be called coacervation. Liposomes, as self-assembled phospholipids structures, were initiated by Bangham mid-1960s [1,2,3,4,5]. Encapsulation as a protective method to stabilize active materials of interest appeared mid-1980s. Active molecules such as drugs, flavours, and perfumes have attracted industrial and scientific attention for encapsulation, particularly with porous-shelled encapsulants [5,6].

Nowadays, encapsulation involves the coating or entrapment of a core material (liquid, solid, or gas) such as food ingredients, enzyme cells, bioactives, or others [7]. Encapsulation can help compounds incorporate into other matrices (e.g., flavours) [8,9], controlled delivery (e.g., drugs) [10,11], or compound stabilization during storage (e.g., antioxidants) [12]. Encapsulation technology is applied in cosmetic, food, pharmaceutical, agriculture, and textile industries due to economic interest in the technology [9,13,14,15,16]. Microencapsulation permits to generate small particles with a wide range of sizes from 1 µm to 1 mm. Spherification process is an encapsulation technique that generates capsules larger than 1 mm (macrocapsules or spheres). Further, capsules with sizes lower than 1 µm are usually called nanocapsules [17,18]. 

The process of alginate gelation is commonly used in encapsulation to create small, ideally spherical particles known as capsules [19]. Alginate, which is generally recognized as safe (GRAS), is often used in this process due to its non-toxic, non-antigenic, biocompatible, and biodegradable properties. Alginate is a versatile material with numerous applications in industries such as textiles, cosmetics, pharmaceuticals, and food, where it is used as a thickener, stabilizer, coating material, and disintegrating agent [20]. Alginate’s ability to form a hydrogel through ionic cross-links with divalent cations, usually calcium ions, makes it a popular choice for encapsulation technology. Alginate is composed of β-D-mannuronic acid (M blocks) and α-L-guluronic acid (G blocks), linked through glycosidic linkage, with the percentage of these blocks varying among different algae species [21]. Alginate gelation is influenced by the size of the polymeric chains, with higher molecular weight leading to tighter cross-linking and higher viscosity gels. Gels with a high percentage of M units are soft and elastic, while those with a high percentage of G units are more compact and rigid [22]. Beads and capsules made from low molecular weight alginate with high G content are compact and rigid [22]. Alginate concentration also affects the gelation process, with higher concentrations resulting in stronger gels and also high-viscosity gels. This aspect should be considered and adapted to the nozzle size used to avoid clogging [17]. It also appears that high-viscosity gels tend to produce non-spherical capsules. In addition, excessive concentration can limit efficient binding opportunities [17].

Alginate is a highly suitable encapsulation material for a wide range of processes. However, to reinforce the functional properties of alginate, numerous studies have explored modifying alginate structure, including using acid hydrolysis, microwave-assisted acidic hydrolysis, enzymatic treatment, thermal treatment, and ultrasound treatment [23,24]. In addition, researchers are investigating composite alginate gels made with other polysaccharides and proteins. These composite gels aim to enhance alginate encapsulation properties, such as encapsulation efficiency, payload, storage stability, and barrier properties [25,26,27]. 

Many encapsulation methods can be used to face the wide range of core and shell materials and targeted capsule size. Spray-drying technology is probably one of the most used encapsulation techniques due to its high productivity but presents limitations for heat-sensitive compounds [28]. Extrusion methods and liposomes are also largely used for encapsulation [29,30]. Coextrusion can be applied to solve limitations encountered for core-shell capsule production by simple extrusion [31], such as size higher than 1 mm, capsule agglomeration, or shell thickness nonuniformity [28,32]. 

In the present paper, a state of the art of coextrusion technology is established. Regrouping the main coextrusion methods used core-shell capsule production. In this study, we detailed their principles and functioning to facilitate further applications.

## 2. Principle of Coextrusion Process 

The target capsules’ size often determines the coextrusion process and the appropriate parameters as liquid feed rate and nozzle diameter (Figure 1). It appears that larger capsule sizes for large nozzle internal diameters are produced at low and high feed rates by dripping (drop-by-drop) and jetting techniques, respectively. In addition, a spray can be obtained by adjusting the nozzle size/feed rate for a reduced capsule size. Based on that, our review will consider four principal methods for encapsulation by coextrusion: dripping system, jetting system, centrifugal system, and electrodynamic system. Additionally, solutions characteristics should be considered before choosing a coextrusion method, especially viscosities, as seen in Figure 1. Indeed, some methods require the use of very small nozzles, those that can be easily clogged at high viscosities. Surface tensions are a key parameter for droplet formation and also need to be considered. In addition, especially for methods applying electric charges, volatility and safety aspects should be examined. 

For efficient encapsulation, core and shell solutions must have mechanical properties compatible with the coextrusion process: viscosities under 2000 mPa.s, interfacial tensions between 10 and 72 mN/m, and fluid densities between 0.7 and 1.3 [33]. For core-shell structure formation, the immiscibility of core and shell solution is an important factor to prevent core and shell solutions from mixing before and during shell gelation. Furthermore, the thermal properties of core and shell solutions should be considered to avoid heating of core material if the coextrusion process requires heating, i.e., waxes [29,34]. 

### 2.1. Dripping Coextrusion System

Dripping coextrusion, also called drop-by-drop coextrusion, is the simplest coextrusion method and serves as foundation for many other coextrusion methods. It produces core-shell capsules through simultaneous extrusion of shell material (generally polymer solution, i.e., alginate) into an external nozzle and the core material (aqueous or lipidic formulation containing the substances to encapsulate) in an internal nozzle. Dripping coextrusion system is presented in Figure 2, where gelling bath contains calcium ions as crosslinking agents for alginate.

Dripping coextrusion is a great technology to produce alginate-based capsules for oil or aqueous cores. For this application, this process has been described mathematically [35]. Therefore, the low flow rate results in the formation of a core-shell drop, which stays attached to the injection system until its weight exceeds the capillary force maintaining it on the nozzle. After the drop fall, a part of it stays attached to the injector, so that, when falling from the nozzle with a diameter *D*, the radius of a core-shell drop *r_d_* can be given by the following empiric equation:(1)rdD≃3.5ClD22.81
where 2/2.81 is a constant value and *C_l_* is the capillary length, defined by
(2)$$C_{l}=\sqrt{\frac{\sigma }{\rho g}}$$
where *σ* is the surface tension (N m^−1^), *ρ* the volumic mass of the liquid (kg m^−3^), and *g* the gravitational force equivalent (9.81 N) [36]. The capillary length is of the order of 2 mm for most liquids [37].

The shell thickness, *h*, of the capsule can be controlled with the ratio between internal and external flow rates [38]. *h* can be approximated from Equation (3):(3)$$h=r_{d}\left(1-{\left(\frac{R_{q}}{1+R_{q}}\right)}^{\frac{1}{3}}\right)$$
with *R_q_* as the flow rates ratio given as
(4)$$R_{q}=\frac{Q_{i}}{Q_{e}}$$
where *Q_i_* is the internal flow rate (core), and *Q_e_* is the external flow rate (shell). 

Experimentally, the shell thickness is not homogenous and is usually thicker at the breaking point with the injector (see Figure 3B). Equation (3) shows that an increase in the ratio *R_q_* induces a decrease in the average shell thickness [36]. The core-centricity probably depends on core solution properties, in addition to the core flow rate. Experimentally, it was observed that small capsules have a more centered core. Additionally, the disparity in core and shell solution densities could generate an off-centered core in core-shell capsules [33].

During dripping, the core-shell drop is characterized by the surface tension related to the shell solution as well as the core solution. For miscible phases, a mix happens inside the drop before gelation and no core-shell capsule can be produced. In addition, when the core-shell drop falls it hits the gelling bath surface leading to the deformation of the capsule. However, surfactant addition to the shell solution as well as in the gelling bath exhibited great results in terms of capsule formation and facilitated drop penetration into gelling solution. Surfactants allowed the production of capsules with thinner shells than without surfactants [39]. 

Capsules properties, such as shape, size, and stability, depend on experimental conditions in addition to the alginate composition, i.e., high-viscosity alginate solutions tend to produce more spherical capsules [17,40] 

To overcome the diffusion phenomenon and fast release of core content, multi-layer capsules can be considered. Dripping coextrusion has been adapted with the addition of a third coaxial tube to allow multi-layered capsules [36,41]. Different shells and core thicknesses can be controlled with core and shell solutions flow rates. An increase of core flow rate induces capsule and core diameter increase, with a decrease of both outer and inner shell thickness. Increasing the intermediate layer flow rate leads to an increase in capsule diameter, an increase in inner shell thickness, and a decrease in outer shell thickness. An increase in the outer layer flow rate results in an increase in capsules’ diameters and outer shell thickness without a significant effect on inner shell thickness and core diameter [41,42]. To reduce and/or avoid the diffusion problem, a lipophilic phase can be injected as an intermediate layer to protect the core phase from core diffusion through the outer shell. As for simple dripping coextrusion, capsule diameters and layer thickness can be predicted. However, this method stays challenging, especially for overcoming coalescence between inner layers, when they are composed of a hydrophilic core and a lipophilic intermediate layer [36]. The triphasic system is also used to facilitate capsules’ formation with an injection of a carrier stream within the outer tube, i.e., polyvinyl alcohol solution. It has shown great results for size and shell thickness control by using a vibrating system coextrusion [43].

### 2.2. Vibrating Coextrusion System 

A vibrating system or annular jet-breaking method is used to produce a capsule diameter smaller than the capillary length (around 1 mm), which was not possible with the dripping method (see Equation (1)) [34,44]. Shell-core capsules are obtained after the injection of the shell material, i.e., alginate solution, and a core material, i.e., oil, simultaneously through a coaxial nozzle. A concentric jet is then produced and separated into droplets under Rayleigh–Plateau instability using a vibrating system. The system is often reinforced with an electrical field with the addition of an electrode ring at the outlet of the injector to create droplets’ electro-repulsion. This artificial stimulation for droplet formation enables sooner formation; closer to the nozzle for smaller droplets [45]. The droplets produced are then collected in a gelling bath [18]. The electro-vibrating system is illustrated in Figure 4.

Experimentally, a solution injection at low flow rates induces a drop-by-drop flowing and generates large capsules, while an increase in the system flow rate (*Q*) generates smaller droplets through a jet formation (*Q_jet_*). The annular jet relies on Rayleigh’s theory of liquid jet instability in order to break it up into droplets [33,45]. Axisymmetric jets lead to droplet formation for conditions described in the following equation: (5)$$Q_{axisymmetric}<81\pi \frac{D^{1.14}{µ}^{0.72}\sigma^{0.14}}{\rho^{0.86}}$$
where *D* is the nozzle diameter, µ is the solution viscosity, *σ* is the surface tension, and *ρ* is the solution density. However, with the increase of flow rate (*Q_max_*), jet length increases until surface instabilities are too high to produce axisymmetric breakup. Therefore, flow rate should be between *Q_jet_* and *Q_max_*. Beyond *Q_max_*, jet instabilities are too great to produce symmetric breakup and correct capsules. An additional increase of the flow rate possibly generates spray (*Q_spray_*) as illustrated in Figure 5, depending on experimental conditions, i.e., solution viscosity [33,46].

The polymer extruded through the injector must have a flow rate (*Q*) high enough to form a continuous and stable jet in order to overcome the effects of viscosity and surface tension. However, an excessively high flow rate (*Q_Spray_* < *Q* < *Q_max_*) would generate a jet high velocity that results in increasing capsules’ impact forces on the gelling bath surface and capsule deformation. This deformation can be limited by decreasing jet velocity or reducing the distance between the nozzle and the gelling bath surface. Such a high velocity could also increase coalescence between droplets and generates doublets capsules (Figure 3). The phenomenon results from the droplets’ velocity fluctuations related to the liquids’ viscoelastic properties; small velocity fluctuations are amplified by capillary and viscous forces acting along the jet. [18,47]. Surfactant addition, i.e., Tween 80, can reduce surface tension and decrease deformation, and also reduce the coalescence phenomenon [18]. 

The transition from dripping mode to jet formation results from an increase in pressure inside the injection system. This transition is known as the dripping–jetting transition (D-J transition) and it is controlled by a Weber number (ratio of inertial to surface tension force) [48,49] given by
(6)$$W_{e}=\frac{\rho Dv^{2}}{\sigma }$$
where *ρ* is the solution volumetric mass, *D* is the nozzle diameter, *v* is the jet velocity relative to the nozzle, and *σ* is the surface tension between the solution and the air. From this, at *W_e_* << 1 the dripping system is observed while at *W_e_* >> 1 jetting is observed. For intermediate values, a dripping faucet system can be observed resulting in a non-homogenous dripping system with interruptions and variable capsule sizes [34].

However, during the jetting–dripping transition, a phenomenon of hysteresis can be observed, probably due to wetting conditions and solution viscosity, but this should be studied further [49]. In addition, solutions with a high elastic module present a particular system, known as the “gobbling phenomenon”. The jet is maintained and the drop continues to grow at the jet end until the maximum droplet size is reached. 

When the force exerted on the jet becomes too great, the drop falls. With polymers solution, and due to the elasticity of the solution, two Weber numbers can be considered, the first *W_e_* beyond the gobbling phenomenon and the second at the transition to jetting [34].

Further, a jet breakup can result in polydisperse droplet size due to jet inconsistencies, and more; satellite particles can be observed. Vibration frequency should match droplet formation to result in monodisperse particle size distributions with d_max_/d_min_ value down to 1.01. The positive effects of the vibrating system for jet breaking on capsule size distribution are shown in Figure 5. 

In order to monitor and adapt the production, the radius of resulting droplets can be predicted by Equation (7):(7)$$r_{d}={\left(\frac{3r_{j}^{2}v_{j}}{4f}\right)}^{1/3}$$
where *r_d_* is the droplet radius, *r_j_* is the jet radius, *v_j_* is the jet linear velocity, and *f* is the frequency of vibration [44,50].

High frequencies increase the coalescence phenomenon, while lower frequencies result in the formation of small droplets (satellites). Satellites can collide with larger droplets during jet-breaking and/or during gelation. Thus, satellites could produce an independent core adjacent to the main one (Figure 3C) or join the core material. The addition of an electrostatic voltage system can reduce the coalescence phenomenon by inducing string polymer negative charges on droplet surfaces, causing them to repel each other [18]. Higher velocities, droplets diameters, and viscosities require higher electrical potential. Charges up to 2–2.15 kV can be applied to the droplets; higher charges could generate their instability [18].

As for capsule size, nozzle dimensions also impact jet formation, so that capsules are generally 1.8 times larger than nozzle diameter, i.e., 900 µm capsules can be expected from a 500 µm nozzle. This is probably due to the jet swelling at the nozzle outlet. The swelling is related to the flow rate; for low flow rates, the swelling occurs immediately at the nozzle outlet, while an increase in flow rate delays the phenomenon as observed in Figure 6 [33,34]. In addition, multiple core phenomena seemed to occur at very high shell and core flow rates [51].

Understanding the mechanisms of capsule formation and the effects of different factors on it improves the adaptability of annular jet coextrusion. It enlarges applications, especially with such a recent encapsulation process. However, the technology’s simplicity makes it a perfect candidate for scale-up. Production rates up to 1000 L/hour can be reached [44]. 

Vibrating coextrusion systems are used in many industries. Nevertheless, the vibrating system presents some limitations. Capsules with a size below 150 µm are complicated to reach and require smaller nozzles. However, decreasing nozzle size enhances probabilities of nozzle plugging, especially with viscous materials. Production of capsules with a range size of 30–8000 µm is then possible with frequency and nozzle size modification [33,52]. 

Various nomenclatures are used in the encapsulation field, therefore, electro-vibrating coextrusion, as well as dripping coextrusion, are sometimes considered as electrospraying methods, when an electrical field is applied for capsule dispersion. In addition, depending on nozzle size, applied voltage, and vibrating frequency, electro-vibrating coextrusion can produce millimetric capsules (spheres) or microcapsules; it is then considered as a spherification or microencapsulation method [33,53].

### 2.3. Centrifugal Coextrusion System

Centrifugal or spinning disc coextrusion consists of the injection of core and shell solution through the concentric orifices located on a rotating cylinder’s outer circumference. As for dripping and vibrating systems, core solutions are pumped separately and injected through an internal tube, and shell solutions through an external tube. As the cylinder rotates, solutions are coextruded and shell fluid sheaths the core solution (Figure 7). Cylinder rotations can operate at a speed in the range of 100 to 1000 rpm. Core-shell drops are formed under centrifugal forces and Rayleigh instabilities due to surface tension. Core-shell capsules are then collected in a gelling bath (alginate gelation) or in another collecting system, i.e., a starch-moving bed [54]. 

As for dripping and vibrating system coextrusion, capsule size is controlled by the nozzles’ size, so that capsules with a diameter ranging from 150 to 2000 µm can be produced. Core-shell capsules’ size is also controlled by the cylinder rotational speed and capsule size decreases with the increase in rotational speed [33,44]. Other parameters should also be considered for size control: solutions’ total flow rate, core and shell flow rates ratio, material properties, and other experimental conditions surrounding the centrifugal cylinder, such as turbulence or airflow [33,54]. Centrifugal coextrusion generates high production capacities compared to dripping systems, so that a 50-nozzle system has a production rate of up to 500 kg/h (Figure 7B). However, due to nozzle multiplication and rotational system, it has a high space consumption. Nevertheless, this provides capsules with longer flight paths so that capsules do not collide with one another in the collection module. In addition, direct observation of droplet formation is more difficult during injection, so monitoring droplets formation for feedback and adjustments is complicated [33,55]. 

### 2.4. Electrohydrodynamic Coextrusion or Electrospraying 

An Electrohydrodynamic (EHD) coextrusion system, also called electrospraying, is an encapsulation process that uses an electric field to produce droplets. Two materials are simultaneously electrosprayed through a coaxial nozzle (needle). The core solution flows through the central capillary, while the shell solution flows through the annular nozzle. An electrical charge is applied to the material between the injection nozzle and the grounded electrode (2–30 kV) [56]. 

The process depends mainly on shell solutions’ properties (viscosity and conductivity). With sufficient shell solution conductivity, encapsulation of a high-resistivity core solution is possible [44,57]. Thus, in EHD coextrusion, capsule shapes depend on shell solution conductivity. Shell solution is often called the driving liquid, even if EHD coextrusion is also possible using conducting core solution and dielectric shell solution [57,58]. Materials extruded through a coaxial nozzle adopt a meniscus shape under the Coulombic interactions, and, depending on experimental conditions, turn into a conical-like shape (Taylor’s cone) (see Figure 8). Electrical forces overcome surface tension and allow the formed jet to break up into highly charged droplets of aerosol [44,56]. Depending on process conditions, electrospraying can be classified into (i) dripping mode or (ii) cone-jet mode (Taylor core mode). Droplets’ generation and size are controlled by process parameters (i.e., feed flow rates, applied voltage, and temperature) and materials characteristics (i.e., solutions viscosity, surface tension, and fluid properties) [59]. Unfortunately, droplets’ size distribution is generally bimodal or multimodal. Generally, primary droplets are the main ones, satellite droplets are formed from instabilities, and offspring droplets are formed from Coulomb fission [60]. However, monodispersed capsules can be obtained by exploiting spray spatial expansion and by adjusting process and materials parameters [61]. 

Microcapsules made with EHD coextrusion have a wide range of sizes. However, compared to dripping coextrusion, EHD coextrusion generates smaller capsules in the range of 1–100 µm. Microcapsules’ diameter can be predicted using the Fernandez de La Mora and Loscertales law, as described by Equation (8):(8)$$d=\alpha {\left(\frac{Q\varepsilon_{0}\varepsilon_{r}}{K}\right)}^{1/3}$$
were *d* is the capsule diameter, *α* is a constant, *Q* is the material flow rate, *ε*_0_ is the permittivity of the free space, *ε_r_* is the relative permittivity of the material, and *K* is the material conductivity [44,63]. The EHD coextrusion process is promising for microcapsules production with high encapsulation efficiency and loading capacity in many fields [57]. However, droplets free of materials can be observed when the shell solutions’ flow rate is too high. On the contrary, when the core solution flow rate is too high compared to the shell solutions’ flow rate, core material droplets can be observed with the shell [57]. 

Nevertheless, the electrohydrodynamic coextrusion process has some challenging limitations, one of them is the low throughput, especially for very small microcapsules production (1–10 µm). As for previous systems, nozzle multiplication has shown efficiency for production rate enhancement and scale-up. Moreover, an arrangement in flow rates ratio with a shell solution rate higher than the core solution rate showed effective encapsulation. The safety aspect is another challenging point concerning the EHD process since the application of high voltages presents major safety concerns, especially in the presence of fine particles [44,57]. 

### 2.5. Other Coextrusion Systems

Other coextrusion systems exist with different modules used to cut annular jets into droplets. To cite some, the jet-cutting technique uses a rotating cutting tool equipped with several wires that cut the jet into smaller cylinders that take the form of spherical droplets under solution surface tension. For the jet-cutting method, capsule size is controlled by the cutting tool rotating frequency, the number of wires on the cutting tool, the wires’ diameter, solutions flow rate, and nozzle design [44,64]. However, capsules smaller than 100 µm are difficult to produce, capsules with sizes in the range of 150 µm to several millimeters have been reported. The method also presents cutting losses produced during the jet cutting which reduced production yields. Those losses can be reduced by recycling and adjusting the cutting tool angle. Unlike annular jet-breaking coextrusion, the jet-cutting process can be used to produce capsules with high-viscosity feed materials. Production yields can be enhanced with multi-nozzle systems [44]. 

The submerged nozzle coextrusion method is also used for core-shell capsule production. Materials are injected through a coaxial nozzle which is submerged within a third concentric nozzle containing the carrier fluid (as illustrated in Figure 9). The nozzle can also be submerged directly into the gelling bath [33,65]. Precipitation and carrier fluid must be immiscible and non-reactive with core and shell solutions for the capsules’ correct formation. The submerged nozzle method generates regular capsules with monodispersed size. In addition, using carrier fluids prevents capsule collisions and deformation under surface tension forces [33]. Several materials can be used, i.e., oils, water, enzymes, and cells [65,66,67]. Submerged nozzle coextrusion generates a wide range of capsules size (500 µm to 8 mm) with smaller capsules by using milli-fluidic technology [33,65,67]. 

Recently, research focused on multi-core coextrusion encapsulation. Some innovative methods have been developed to produce capsules with more than one core, allowing the encapsulation of different ingredients into separated cores. The number of cores can be controlled by the number of inner tubes [69]. Multi-core capsules have been successfully used for α-tocopherol encapsulation. This also presented great results for probiotics encapsulation, where *lactobacillus* and *Bacillus subtilis* have been encapsulated into separated cores without affecting bacterial proliferation [70]. Qu et al. (2022) reported a very promising protocol for producing a eukaryotic cell-like system, i.e., multicompartmental capsules with distinct enzymatic cellular reactions (Figure 10) [71]. Another approach to producing multicompartmental capsules proposed by He et al. (2016) consists of the combination of two or three coaxial nozzles by placing them near each other vertically, the system is presented Figure 10. Thus, the generated droplets could merge before hitting the gelling bath and form compartmented capsules [72]. High application potential is promised by the multi-core and the multicompartmental capsules for coencapsulation applications such as drug codelivery, confined reactions, enzyme immobilizations, and cell cultures [71]. 

## 3. Advantages and Limitations of Coextrusion Methods

All the coextrusion methods offer advantages such as a protective barrier for encapsulated ingredients, slow-release kinetics, and stability against oxidation. However, there are limitations such as process dependency on core and shell fluid rates and the requirement for high-precision equipment [17,29,73,74,75]. Nevertheless, this paper described four coextrusion methods and each has interests and weaknesses. The dripping method produces multilayered and large capsules but has a non-homogeneous shell thickness [33,36,41]. The vibrating method produces small capsules with high production rates but requires many trials and has a high probability of nozzle plugging and capsule coalescence. The centrifugal method allows for high production capacities but, on the other hand, requires a large space and direct monitoring, which can be difficult [18,33,47]. In addition, EHD produces a wide size range and smaller capsules but has a lower payload and safety concerns due to high voltage use; more advantages and limitations are displayed in Table 1. The choice of encapsulation method depends on the specific requirements of the application, such as the size of the desired capsule, the fluid properties, and the production capacity.

## 4. Conclusions

The encapsulation process aims to enclose one or more substances within a semi-permeable coating material to produce particles, solid beads, or capsules. It is a protective technology that is widely used nowadays.

Coextrusion technology involves a concentric nozzle and an additional system for capsules’ separation (vibration unit, spinning disc, etc.). The technology allows the production of capsules with a wide size range from a few microns to several millimeters [33]. Coextrusion is used for core-shell capsules’ production, with a higher payload than other techniques, such as spray drying. As discussed, there are various coextrusion process variations, i.e., dripping system, vibrating system, centrifugal system, and EHD system. 

The choice of coextrusion technology is, therefore, driven by the droplet formation that depends on a combination of multiple parameters: (i) material properties, i.e., viscosity, interfacial tension, solubility, and density; (ii) nozzle size according to the capsule size expected; (iii) scale-up possibilities; (iv) production rates; and (v) the need of a sterile environment. 

This review showed that coextrusion technology presents several advantages compared to other encapsulation methods, i.e., large payload, continuous process, etc., but also has limitations such as the requirement of high precision equipment and difficulty of process optimization. 

## Figures and Tables

**Figure 1 marinedrugs-21-00235-f001:**
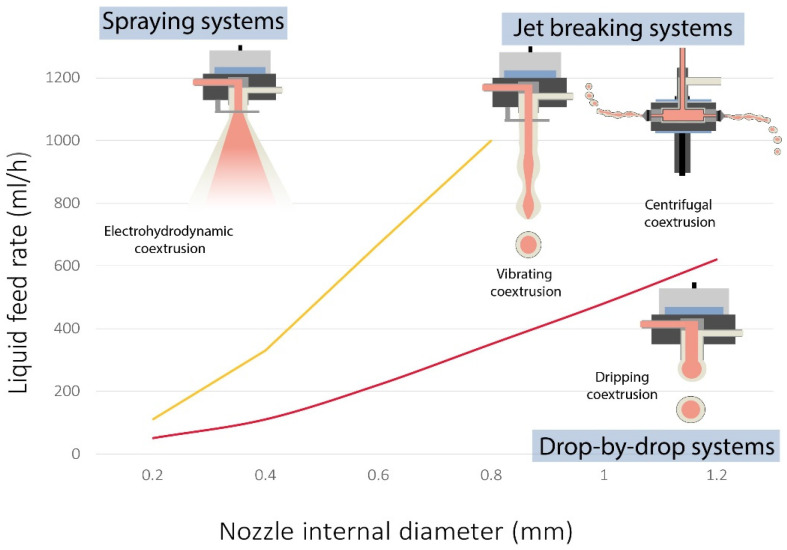
Illustration of different injection methods produced with various feed rates and nozzle internal diameter.

**Figure 2 marinedrugs-21-00235-f002:**
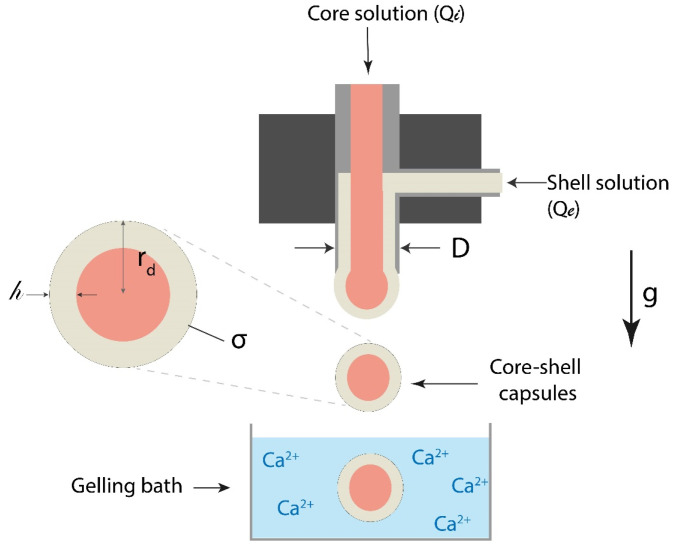
Dripping coextrusion system process for “alginate shell-core” capsules formation.

**Figure 3 marinedrugs-21-00235-f003:**
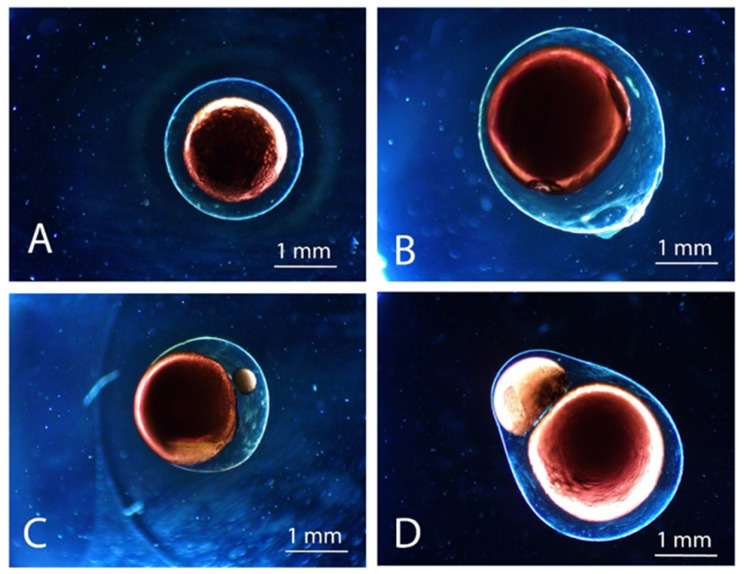
Microscopy pictures of different core-shell capsules’ aspects produced by coextrusion technology: (**A**) centered core, (**B**) decentered core, (**C**) satellite incorporation into the shell, and (**D**) doublet capsule.

**Figure 4 marinedrugs-21-00235-f004:**
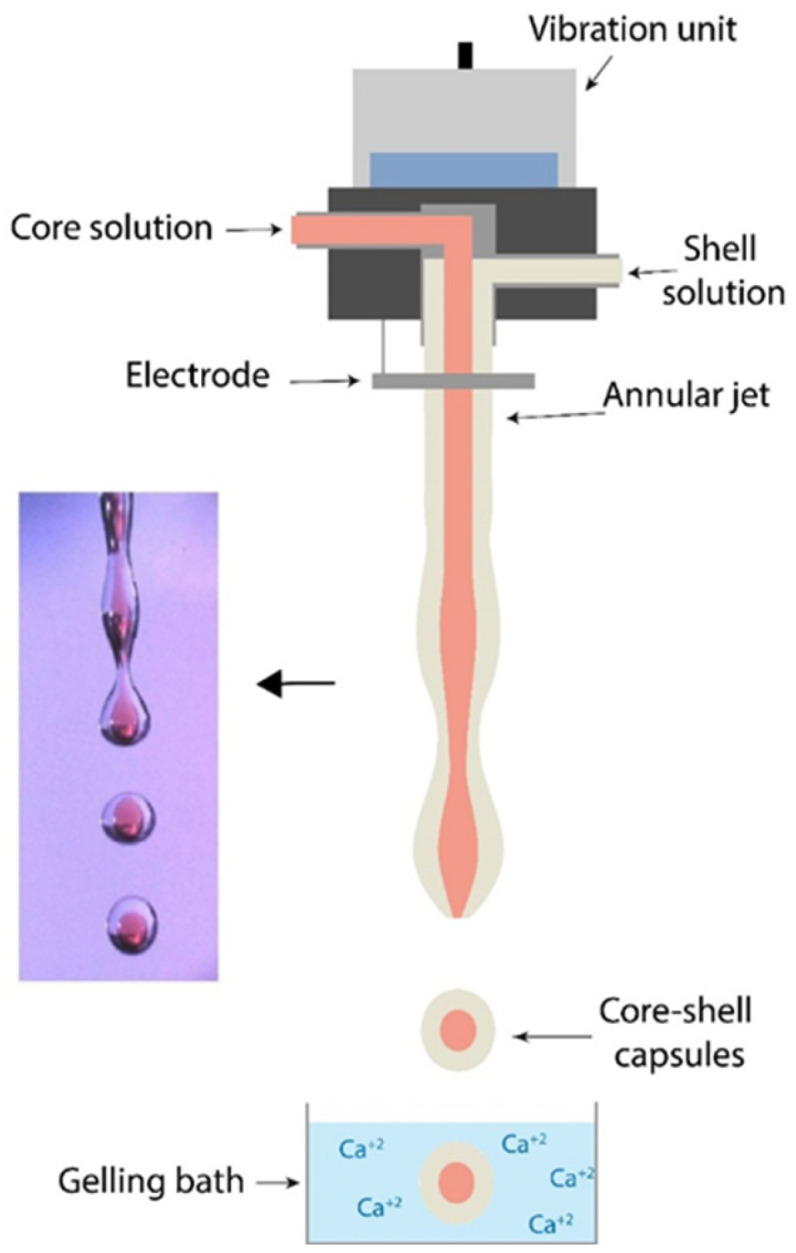
Example of an electro-vibrating coextrusion device for alginate capsules.

**Figure 5 marinedrugs-21-00235-f005:**
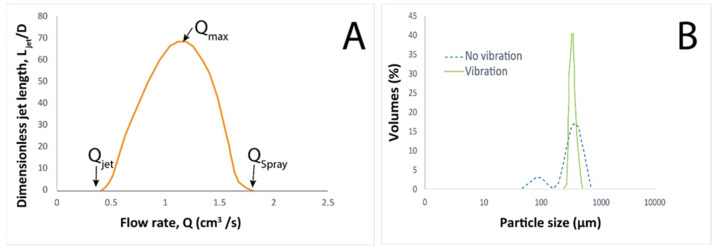
(**A**) Particle size distribution of core-shell microcapsules with and without a vibration system. (**B**) Flow rate influence on jet breakup (Adapted [33]).

**Figure 6 marinedrugs-21-00235-f006:**
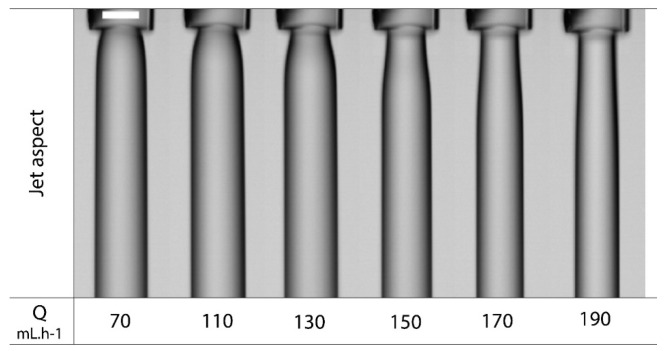
Jet diameter increase at nozzle outlet and its variation with the injection feed rate (Q) (adapted [34]).

**Figure 7 marinedrugs-21-00235-f007:**
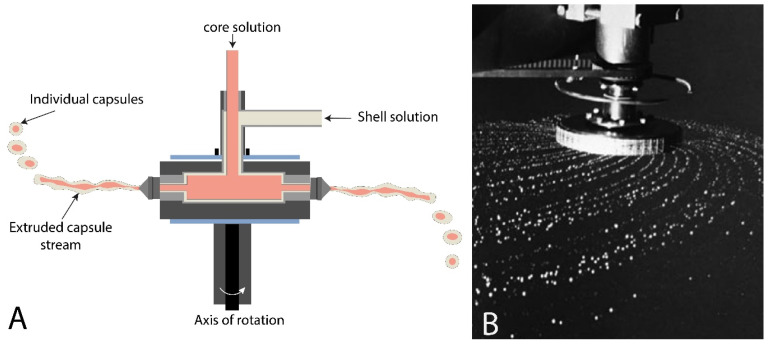
(**A**) Illustration of centrifugal coextrusion and (**B**) photograph of centrifugal coextrusion units with fifty nozzles (Adapted [33]).

**Figure 8 marinedrugs-21-00235-f008:**
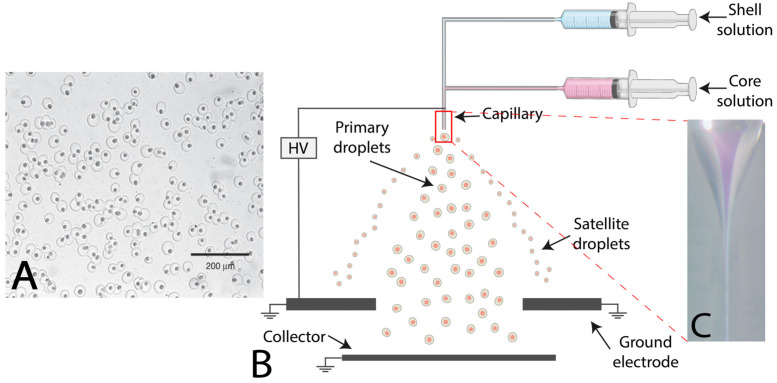
(**A**) Microphotography capsules produced with electrohydrodynamic coextrusion. (**B**) Schematic of electrohydrodynamic (EHD) system setup. (**C**) A close-up view of a coextruded feed stream through a coaxial nozzle forming a Taylor cone (Adapted [62]).

**Figure 9 marinedrugs-21-00235-f009:**
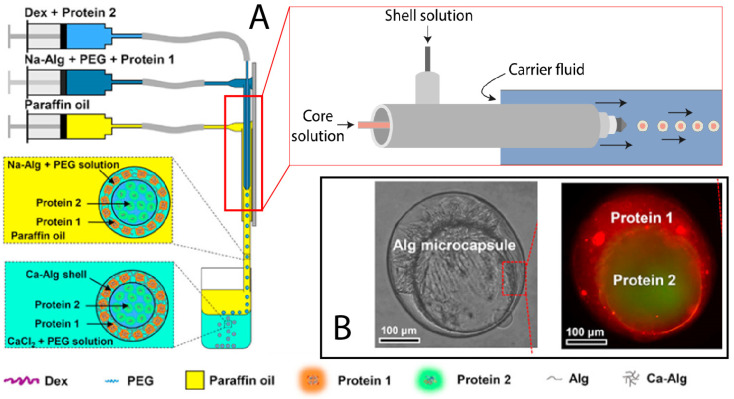
(**A**) Illustration of submerged nozzle coextrusion, (**B**) microphotography of core-shell alginate and proteins capsules obtained by submerged nozzle coextrusion (Adapted [68]).

**Figure 10 marinedrugs-21-00235-f010:**
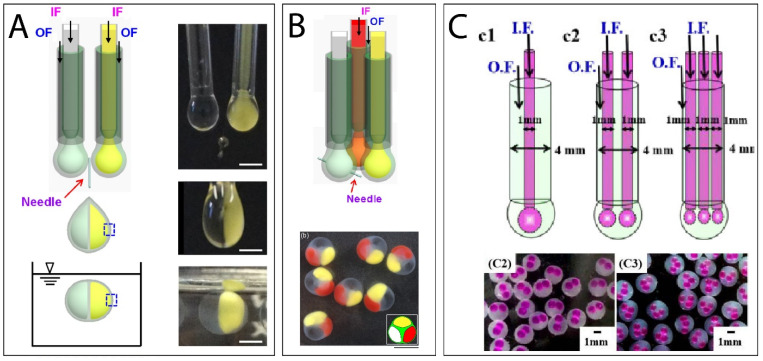
(**A**) Coextrusion system for the production of dual-compartmental capsules, (**B**) coextrusion system for the production of triple-compartmental capsules, and (**C**) coextrusion systems for the production of multi-cores capsules. (Adapted [69,71]). *IF: Internal flow, OF: Outer flow*.

**Table 1 marinedrugs-21-00235-t001:** Advantages and limitations of coextrusion methods for core-shell capsule production.

Methods	Advantages	Limitations	Refs
Dripping system	Scale-up possible through nozzle multiplication Possibility to produce multilayered capsules Production of Large capsules	Non homogenous shell thickness High dependency on fluids surface tensions	[33,36,41]
Vibrating system	Facilities to control size and shell thickness Production of small capsules High production rates Wide range of capsule size	Required many trials to find the appropriate vibration frequency Process not adapted to high viscosities fluids encapsulation Possibility of capsule coalescence Multiple core phenomena when variables are not adapted High nozzle plugging probabilities.	[18,33,47]
Centrifugal system	Possibility to use various collecting systems High production capacities Reduced coalescence	High space consumptions Direct monitoring can be difficult	[33,44,55]
EHD system	Wide size range Possibility to produce smaller capsules Scale-up with nozzle multiplication	Dependence on solutions viscosities and conductivities Bimodal or multimodal size distribution. Low throughput Lower payload Safety aspect, due to high voltage use.	[33,44,57]
Common to all methods	Large payload related to a reservoir-type structure. The shell acts as a protective barrier for encapsulated ingredient protection compared to simple extrusion process. Slow-release kinetics brought by the shell Stability of the encapsulated ingredient against oxidation Limitation of highly volatile actives evaporation (e.g., Essential oils) Burst-like effect obtained by the highly loaded core-shell capsules breakage. (e.g., high flavor release in chewing gum) Continuous process	Process dependency on core and shell fluids rate Hydrophilic liquids encapsulation may be complicated. Many experimental parameters have to be defined for their impact on capsules properties. High precision equipment is required Additional step is required to get powder form	[17,29,73,74,75]

## Data Availability

Data sharing not applicable.

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
