# Peer review of "Alginate Core-Shell Capsules Production through Coextrusion Methods: Principles and Technologies"

_marinedrugs, 2023, doi:10.3390/md21040235_

Round 1

Reviewer 1 Report

In the present work, the authors carried out a review of coextrusion technology and presented the main coextrusion methods used in the production of core-shell capsules. The paper brings a lot of relevant information and will certainly contribute greatly to the development of this area of knowledge. However, some improvements need to be made to the text for it to be publishable.

Suggestions:

1 – Introduction can be improved and enriched with more data from the literature;

2- A new section should be created to show the advantages and limitations of coextrusion methods and a detailed discussion should be done. Table 1 alone is not enough to approach this subject. Furthermore, table 1 is loose in the text and a connection must be created.

These changes must be made for the paper  to be publishable.

Author Response

In the present work, the authors carried out a review of coextrusion technology and presented the main coextrusion methods used in the production of core-shell capsules. The paper brings a lot of relevant information and will certainly contribute greatly to the development of this area of knowledge. However, some improvements need to be made to the text for it to be publishable.

Suggestions:

1 – Introduction can be improved and enriched with more data from the literature;

Authors answer: We would like the thank the reviewer for his valuable time spending on reviewing our paper, and for his suggestions.

Introduction has been enriched with more information about alginate and data from the literature, modifications are highlighted in the manuscript.

“The process of alginate gelation is commonly used in encapsulation to create small, ideally spherical particles known as capsules [19]. Alginate, which is Generally Recognized As Safe (GRAS), is often used in this process due to its non-toxic, non-antigenic, biocompatible, and biodegradable properties. Alginate is a versatile material with numerous applications in industries such as textiles, cosmetics, pharmaceuticals, and food, where it is used as a thickener, stabilizer, coating material, and disintegrating agent [20]. Alginate ability to form a hydrogel through ionic cross-links with divalent cations, usually calcium ions, makes it a popular choice for encapsulation technology. Alginate is composed of β-D-mannuronic acid (M blocks) and α-L-guluronic acid (G blocks), linked through glycosidic linkage, with the percentage of these blocks varying among different algae species [21]. Alginate gelation is influenced by the size of the polymeric chains, with higher molecular weight leading to tighter cross-linking and higher viscosity gels. Gels with a high percentage of M units are soft and elastic, while those with a high percentage of G units are more compact and rigid [22]. Beads and capsules made from low molecular weight alginate with high G content are compact and rigid [22]. Alginate concentration also affects the gelation process, with higher concentrations resulting in stronger gels, but also high viscosity gels. This aspect should be considered and adapted to nozzle size used to avoid clogging [17]. It also appears that high viscosity gels tend to produce non spherical capsules. Also, excessive concentration can limit efficient binding opportunities [17].

Alginate is a highly suitable encapsulation material for a wide range of processes. However, to reinforce the functional properties of alginate, numerous studies have explored modifying alginate structure, including using acid hydrolysis, micro-wave-assisted acidic hydrolysis, enzymatic treatment, thermal treatment, and ultra-sound treatment [23,24]. Also, researchers are investigating composite alginate gels made with other polysaccharides and proteins. These composite gels aim to enhance alginate encapsulation properties, such as encapsulation efficiency, payload, storage stability, and barrier properties [25–27].”

2- A new section should be created to show the advantages and limitations of coextrusion methods and a detailed discussion should be done. Table 1 alone is not enough to approach this subject. Furthermore, table 1 is loose in the text and a connection must be created.

Authors answer:  According to suggestion, a new section has been created discussing advantages and limitations of coextrusion method. The new section is highlighted in the manuscript as follow:

Advantages and limitations of coextrusion methods

All the coextrusion methods offer advantages such as a protective barrier for encapsulated ingredients, slow-release kinetics, and stability against oxidation. However, there are limitations such as process dependency on core and shell fluid rates and the requirement for high precision equipment [17,29,73–75]. Nevertheless, this paper de-scribed four coextrusion methods and each have interests and weaknesses. Dripping method produces multilayered and large capsules, but has a non-homogeneous shell thickness [33,36,41]. Vibrating method produces small capsules with high production rates, but requires many trials and has a high probability of nozzle plugging and capsule coalescence. While centrifugal method allows for high production capacities but on the other side, it requires a large space and direct monitoring can be difficult [18,33,47]. Also, EHD produces a wide size range and smaller capsules but has a lower payload and safety concerns due to high voltage use, more advantages and limitations are displayed in table 1. The choice of encapsulation method depends on the specific requirements of the application, such as the size of the desired capsule, the fluid properties, and the production capacity.”

These changes must be made for the paper to be publishable.

Reviewer 2 Report

Please see the comments in the attached file

Reviewer 3 Report

First of all, congratulations for the good work done.

It is described and structured in a simple and easy to understand way. I really liked the description of the different coextrusion methods, advantages and disadvantages, influencing factors, etc. Table 1 has been very useful to me. Likewise, all the figures have seemed very illustrative.

I would only miss the justification for the choice of alginate in the production of core-shell capsules and why a paragraph on the advantages of this polysaccharide has not been included in the introduction. It is one of the words that appears in the title of the publication and among the keywords. A brief description of the types of alginates used in the production of shells using the coextrusion process and how the viscosity of the different alginates influences their formation would be nice.

For the rest, I consider that the work is perfect for publication.

Author Response

First of all, congratulations for the good work done.

It is described and structured in a simple and easy to understand way. I really liked the description of the different coextrusion methods, advantages and disadvantages, influencing factors, etc. Table 1 has been very useful to me. Likewise, all the figures have seemed very illustrative.

I would only miss the justification for the choice of alginate in the production of core-shell capsules and why a paragraph on the advantages of this polysaccharide has not been included in the introduction. It is one of the words that appears in the title of the publication and among the keywords. A brief description of the types of alginates used in the production of shells using the coextrusion process and how the viscosity of the different alginates influences their formation would be nice.

For the rest, I consider that the work is perfect for publication.

Authors answer:  Authors thank the reviewer for his valuable time. Suggested modifications have been made in the introduction and they are lighted in the manuscript as follow:

“The process of alginate gelation is commonly used in encapsulation to create small, ideally spherical particles known as capsules [19]. Alginate, which is Generally Recognized As Safe (GRAS), is often used in this process due to its non-toxic, non-antigenic, biocompatible, and biodegradable properties. Alginate is a versatile material with numerous applications in industries such as textiles, cosmetics, pharmaceuticals, and food, where it is used as a thickener, stabilizer, coating material, and disintegrating agent [20]. Alginate ability to form a hydrogel through ionic cross-links with divalent cations, usually calcium ions, makes it a popular choice for encapsulation technology. Alginate is composed of β-D-mannuronic acid (M blocks) and α-L-guluronic acid (G blocks), linked through glycosidic linkage, with the percentage of these blocks varying among different algae species [21]. Alginate gelation is influenced by the size of the polymeric chains, with higher molecular weight leading to tighter cross-linking and higher viscosity gels. Gels with a high percentage of M units are soft and elastic, while those with a high percentage of G units are more compact and rigid [22]. Beads and capsules made from low molecular weight alginate with high G content are compact and rigid [22]. Alginate concentration also affects the gelation process, with higher concentrations resulting in stronger gels, but also high viscosity gels. This aspect should be considered and adapted to nozzle size used to avoid clogging [17]. It also appears that high viscosity gels tend to produce non spherical capsules. Also, excessive concentration can limit efficient binding opportunities [17].

Alginate is a highly suitable encapsulation material for a wide range of processes. However, to reinforce the functional properties of alginate, numerous studies have explored modifying alginate structure, including using acid hydrolysis, micro-wave-assisted acidic hydrolysis, enzymatic treatment, thermal treatment, and ultra-sound treatment [23,24]. Also, researchers are investigating composite alginate gels made with other polysaccharides and proteins. These composite gels aim to enhance alginate encapsulation properties, such as encapsulation efficiency, payload, storage stability, and barrier properties [25–27].”